# Characterising COVID-19 empirical research production in Latin America and the Caribbean: A scoping review

Cristián Mansilla[1]☯*, Cristian A. Herrera[2,3]☯*, Laura Boeira[4], Andrea Yearwood[5], Analia S. Lopez[6], Luis E. Colunga-Lozano[7], Eva Brocard[8], Tatiana Villacres[9], Marcela Vélez[10], Gabriel Di Paolantonio[11], Ludovic Reveiz[12]

1 McMaster Health Forum and Health Policy PhD Program, McMaster University, Hamilton, Canada,
2 Department of Public Health, School of Medicine, Pontificia Universidad Católica de Chile, Santiago, Chile,
3 The World Bank Group, Washington, DC, United States of America, 4 Instituto Veredas, Porto Alegre, Brazil, 5 Independent Researcher, Port of Spain, Trinidad and Tobago, 6 Instituto Universitario CEMIC (IUC), Buenos Aires, Argentina, 7 Department of Clinical Medicine, Universidad de Guadalajara, Guadalajara, Jalisco, Mexico, 8 Independent Researcher, Paris, France, 9 Quantics, Quito, Ecuador, 10 Faculty of Medicine, Universidad de Antioquia, Medellin, Colombia, 11 PhD in Economics Program, University of Paris 1, Paris, France, 12 Incident Management Systems for COVID-19 and Evidence and Intelligence for Action in Health Department, Pan American Health Organization, Washington, DC, United States of America

☯ These authors contributed equally to this work.
* camansil@gmail.com (CM); crherrer@uc.cl (CAH)

## Abstract

### Introduction

The Coronavirus Disease 2019 (COVID19) pandemic has struck Latin America and the Caribbean (LAC) particularly hard. One of the crucial areas in the international community's response relates to accelerating research and knowledge sharing. The aim of this article is to map and characterise the existing empirical research related to COVID-19 in LAC countries and contribute to identify opportunities for strengthening future research.

### Methods

In this scoping review, articles published between December 2019 and 11 November 2020 were selected if they included an empirical component (explicit scientific methods to collect and analyse primary data), LAC population was researched, and the research was about the COVID-19 pandemic, regardless of publication status or language. MEDLINE, EMBASE, LILACS, Scielo, CENTRAL and Epistemonikos were searched. All titles and abstracts, and full texts were screened by two independent reviewers. Data from included studies was extracted by one reviewer and checked by a second independent reviewer.

### Results

14,406 records were found. After removing duplicates, 5,458 titles and abstracts were screened, of which 2,323 full texts were revised to finally include 1,626 empirical studies. The largest portion of research came from people/population of Brazil (54.6%), Mexico

**Data Availability Statement:** All relevant data are within the manuscript and its Supporting Information files.

**Funding:** The author(s) received no specific funding for this work.

**Competing interests:** The authors have declared that no competing interests exist. Authors hold sole responsibility for the views expressed in the manuscript, which may not necessarily reflect the opinion or policy of the Pan American Health Organization nor the World Bank Group.

(19.1%), Colombia (11.2%), Argentina (10.4%), Peru (10.3%) and Chile (10%), while Caribbean countries concentrated 15.3%. The methodologies most used were cross-sectional studies (34.7%), simulation models (17.5%) and randomized controlled trials (RCTs) (13.6%). Using a modified version of WHO's COVID-19 Coordinated Global Research Roadmap classification, 54.2% were epidemiological studies, followed by clinical management (22.3%) and candidate therapeutics (12.2%). Government and public funds support were reported in 19.2% of studies, followed by universities or research centres (9%), but 47.5% did not include any funding statement.

## Conclusion

During the first part of the COVID-19 pandemic, LAC countries have contributed to the global research effort primarily with epidemiological studies, with little participation on vaccines research, meaning that this type of knowledge would be imported from elsewhere. Research agendas could be further coordinated aiming to enhance shared self-sufficiency regarding knowledge needs in the region.

## Introduction

The Coronavirus Disease 2019 (COVID-19) pandemic has struck Latin America and the Caribbean (LAC) particularly hard, having substantial health, social and economic consequences for the population living in this Region. By the end of October 2021, the SARS-CoV-2 has infected over 46 million people (almost 19% of all infections in the world) and caused over 1.5 million deaths (more than 30% of all registered deaths globally) [1], while the region represents less than 8.5% of the world population [2], along with an estimated reduction of 7.4% in gross domestic product (GDP), only in 2020 [3]. The pandemic has importantly impacted LAC countries, in a context of high levels of inequality and labour informality than other regions, with comparatively weaker social protection schemes, along with health systems feebly prepared to boost test, track and trace programmes and to face high demand surges for specialised intensive care, especially when compared with high-income countries and even some middle-income countries [4].

At the global level, one of the crucial areas in the international community's response to COVID-19 relates to accelerating research, innovation and knowledge translation and sharing [5]. On 11 and 12 February 2020, the Global Research Forum, hosted by WHO, developed an initial COVID-19 Global Research Roadmap with two main aims: "1. to facilitate that those affected are promptly diagnosed and receive optimal care; while integrating innovation fully within each research area; and 2. to support research priorities that will lead to the development of sustainable global research platforms that are prepared for the next disease X epidemic" [6]. Back then, this Forum identified groups actively researching on COVID19 in Africa, Australia, Europe and North America, but no mention was made of LAC.

Despite these global efforts to encourage research collaboration across countries and regions, it is not known if the research that has been conducted in LAC in response to the COVID-19 crisis accounts for the impact that the pandemic is having in the region. A literature review conducted in April 2020 found that only 2.7% of the total publications related to COVID-19 had at least one author with a Latin American-based affiliation [7], showing some insights of the relatively low development of research in LAC at that time. This was

corroborated with subsequent bibliometric studies showing the same low participation of LAC [8, 9]. A more recent report of the Organisation for Economic Cooperation and Development [10] analysed 74 115 COVID-19 documents in PubMed during the period 1 January to 30 November 2020, finding that in terms of author's affiliation the United States represented 23% and the European Union 22% of all documents, followed by China, the United Kingdom and India. The top collaboration partnerships also occurred between these countries, signaling that the LAC region has not significantly participated in COVID-19 research production.

Additionally, understanding the development of research during the COVID-19 pandemic becomes especially relevant and necessary as we have seen a "coviditisation" of research [11]. This has brought challenges related to redundancy and research waste, leaving other relevant fields unattended, or with a diminished research quality. For instance, a paper evaluating the characteristics and expected strength of evidence of COVID-19 studies registered on Clinical-Trials.gov found that only 29.1% have the potential to result in OCEBM level 2 evidence (good-quality evidence) and that of the randomised clinical trials protocols, only 29.3% are placebo-controlled, blinded studies [12]. More recently, we learned that researchers registered more than 10,000 clinical trials related to COVID-19, but the majority were too small or poorly designed, and in some cases there have been an excess of trials for some particular interventions (e.g. hydroxychloroquine) [13]. In addition, research funding has also been questioned in terms of the distribution and sources for achieving a fair and sustainable research and development environment [14]. In LAC, a study searching for COVID-19 trials on treatment and prevention in the region identified "a trend towards small repetitive non-rigorous studies that duplicate efforts and drain limited resources without producing meaningful conclusions on the safety and efficacy of the interventions being tested" [15]. However, apart from this study it is unknown what are the trends of empirical research in LAC during the COVID-19 pandemic, in a context where institutional development for health research has progressed in the region but has been reported as generally uncoordinated and disaggregated, and uneven between countries [16].

Considering the large impact that COVID-19 has had in LAC and the relevance of research for the COVID-19 response, there is a need to better understand how research production has unfolded in the region to inform contextually relevant decision making. The aim of this article is to map and characterise the existing empirical research related to COVID-19 in LAC countries during the pandemic and contribute to identify opportunities for strengthening research for the future.

## Methods

This is a scoping review of the existing empirical research produced in LAC countries related to COVID-19 pandemic. The Preferred Reporting Items for Systematic reviews and Meta-Analysis for scoping reviews (PRISMA-ScR) checklist can be found in the Supporting information.

### Eligibility criteria

Articles in any language were eligible. To be included:

- articles needed to be empirical research demonstrated by the report of the scientific methods used to collect and analyze the primary data;

- the population being researched must include people from at least one LAC country or their explicit focus is on one or more LAC countries. LAC countries considered are Antigua and Barbuda, Argentina, Bahamas, Barbados, Belize, Bolivia, Brazil, Chile, Colombia, Costa Rica,

Cuba, Dominica, Dominican Republic, Ecuador, El Salvador, Grenada, Guatemala, Guyana, Haiti, Honduras, Jamaica, Mexico, Nicaragua, Panama, Paraguay, Peru, Saint Kitts and Nevis, Saint Lucia, Saint Vincent and the Grenadines, Suriname, Trinidad and Tobago, Uruguay, and Venezuela;

- the research must be directly or explicitly connected to the current COVID-19 pandemic.

Articles were excluded if they:

- were any type of evidence synthesis of the literature, or documents building on evidence syntheses (e.g., guidelines, recommendations, consensus, systematic reviews, etc.);

- used large international databases and their explicit focus was not in one or more LAC countries;

- were not using empirical data to build their findings (including clinical case report).

Articles were included regardless of their study design, topic and publication date. Simulation studies were also included if they considered empirical data to build on a specific model (e.g., case counts, tests, etc.).

## Search methods

To identify potentially relevant documents, the following bibliographic databases were searched:

- MEDLINE and EMBASE using Ovid (December 2019 to 11 November 2020)

- LILACS (using VHL) (inception to 11 November 2020)

- Scielo (2019 to 11 November 2020)

- CENTRAL (2019 to 11 November 2020)

- Epistemonikos (2019 to 11 November 2020).

To identify grey literature, registry of trials, and pre-print articles, the electronic database search was also supplemented by searching the Living Overview of the Evidence (L-OVE) COVID-19 Repository by the Epistemonikos Foundation, and searching the references of evidence syntheses that were found when assessing the eligibility of the articles. Additionally, registries for studies that are planned to be conducted were complemented with the clinical trials found by Carracedo S et al. 2021 [15]. The search strategies were built based on all potential synonyms of "COVID-19", combining with geographical terms that could point out towards the LAC region, or any of the countries that are part of the region. The full search strategies used in each database are described in the Supporting information.

## Study selection

Duplicates were removed using EndNote® and Covidence®. All title and abstracts, and full texts were screened by two independent reviewers, resolving disagreements by a third reviewer, or a formal discussion between the two involved reviewers. Covidence ® was used to conduct this process.

## Data extraction and charting

From the included articles, data was extracted by one reviewer, and was checked by a second independent reviewer, agreeing on what data to extract from each study. The following

characteristics of the included studies were extracted (see Supporting information for a complete description of each item extracted):

- Lead author, month, year, and citation

- Data sources, classified as inert sources, animals, directly humans, databases or documents.

- Main objective of the article and research question classified as exposure, prevalence or incidence, benefits and harms of an intervention, prognosis, views and preferences, diagnostic accuracy, or other.

- Methods and study design of each study, identifying basic sciences, quantitative, qualitative or mixed-methods studies.

- Type of intervention, exposure, or phenomenon of interest being addressed, using the 9 main research areas outlined by the World Health Organization in their Global Research Roadmap [6], and adding health systems arrangements as one key area. In addition, a "Other" category was added to include topics like the impact of the pandemic in mental health.

- Countries of the LAC region where the research is being targeted.

- Funding of the study, classified as government and public funds, international organizations, universities or research centres, NGOs, private companies, public private entities, or others.

- Journal or website where the study was published.

- Status of publication.

The data extraction template was piloted with 10 studies by two authors, and the full data extraction process was conducted in Microsoft Excel ⓇⓇ.

## Data synthesis

With the information collected from each study, descriptive information is presented to characterize the research that was being conducted on LAC countries.

The total volume of research that was produced by month is presented, and the number of publications per country is analyzed, showing absolute numbers and a rate of publications per population. Rate per the number of researchers in each country is also calculated, based on the information published by UNESCO [17]. The geographical information is also presented in interactive maps using Tableau Ⓡ software (see link provided in Table 1).

Summaries of the main data sources used, methodologies and study designs, type of research questions and sources of funding are also calculated.

Finally, descriptive summaries of the number of publications addressing each one of the modified WHO Global Research Roadmap categories are also presented.

## Results

### Search results

14,406 records were found. After removing duplicates, 5,458 titles and abstracts were screened, of which 2,323 full texts were revised to finally include 1,626 empirical studies. See Fig 1 for the flow diagram of the review. The full list of included studies with all the extracted data from them, along with all the excluded studies with the reasons for exclusion is provided in the Supporting information.

**Timing and status of the publication.** Among the 1,626 publications included in this analysis, the month on which it was available to the public is shown in Fig 2. The number of

**Table 1. Empirical studies related to COVID-19 in each LAC country.**

| Country | Frequency | Population | Rate (publications per 1,000,000 population) | Rate of research per FTE researchers*1000 [17] |
|---|---|---|---|---|
| Antigua and Barbuda | 12 | 97,929 | 122.54 | |
| Argentina | 168 | 45,195,774 | 3.72 | 3.26 |
| Bahamas | 12 | 393,244 | 30.52 | |
| Barbados | 11 | 287,375 | 38.28 | |
| Belize | 15 | 397,628 | 37.72 | |
| Bolivia | 52 | 11,673,021 | 4.45 | 31.59 |
| Brazil | 885 | 212,559,417 | 4.16 | 6.38 |
| Chile | 162 | 19,116,201 | 8.47 | 23.83 |
| Colombia | 181 | 50,882,891 | 3.56 | 23.5 |
| Costa Rica | 41 | 5,094,118 | 8.05 | 6.71 |
| Cuba | 70 | 11,326,616 | 6.18 | |
| Dominica | 9 | 71,986 | 125.02 | |
| Dominican Republic | 41 | 10,847,910 | 3.78 | |
| Ecuador | 106 | 17,643,054 | 6.01 | 38.74 |
| El Salvador | 35 | 6,486,205 | 5.40 | |
| Grenada | 9 | 112,523 | 79.98 | |
| Guatemala | 39 | 17,915,568 | 2.18 | 94.89 |
| Guyana | 11 | 786,552 | 13.99 | |
| Haiti | 16 | 11,402,528 | 1.40 | |
| Honduras | 35 | 9,904,607 | 3.53 | |
| Jamaica | 17 | 2,961,167 | 5.74 | |
| Mexico | 310 | 128,932,753 | 2.40 | 6.72 |
| Nicaragua | 26 | 6,624,554 | 3.92 | |
| Panama | 42 | 4,314,767 | 9.73 | 95.89 |
| Paraguay | 47 | 7,132,538 | 6.59 | 43.48 |
| Peru | 167 | 32,971,854 | 5.06 | |
| Saint Kitts and Nevis | 8 | 53,199 | 150.38 | |
| Saint Lucia | 8 | 183,627 | 43.57 | |
| Saint Vincent and the Grenadines | 7 | 110,940 | 63.10 | |
| Suriname | 10 | 586,632 | 17.05 | |
| Trinidad and Tobago | 13 | 1,399,488 | 9.29 | |
| Uruguay | 52 | 3,473,730 | 14.97 | 28.84 |
| Venezuela, Bolivarian Republic of | 50 | 28,435,940 | 1.76 | 5.76 |

Notes: Blank cells means that the rate of researchers per population was not available for the country.

Figures underlying data can be found in https://public.tableau.com/views/LACCOVIDresearch-scopingreview/publicationspercountry?:language=en-US&publish= yes&:display_count=n&:origin=viz_share_link.

publications had a sharp increase from the start of the pandemic, starting mainly in March and reaching the peak in July. The level kept relatively constant over 200 publications per month between May and September. 5 of the articles did not report the month in which it was available, while 1 of them was retracted from the journal. Most articles were published in scientific journals (886; 55%), followed by pre-print versions (444; 27%) and registries (287; 18%).

## Geographical distribution

In terms of the geographical distribution of the COVID-19 empirical research (see Table 1), all 33 countries in the region had at least one publication conducted in their population. There

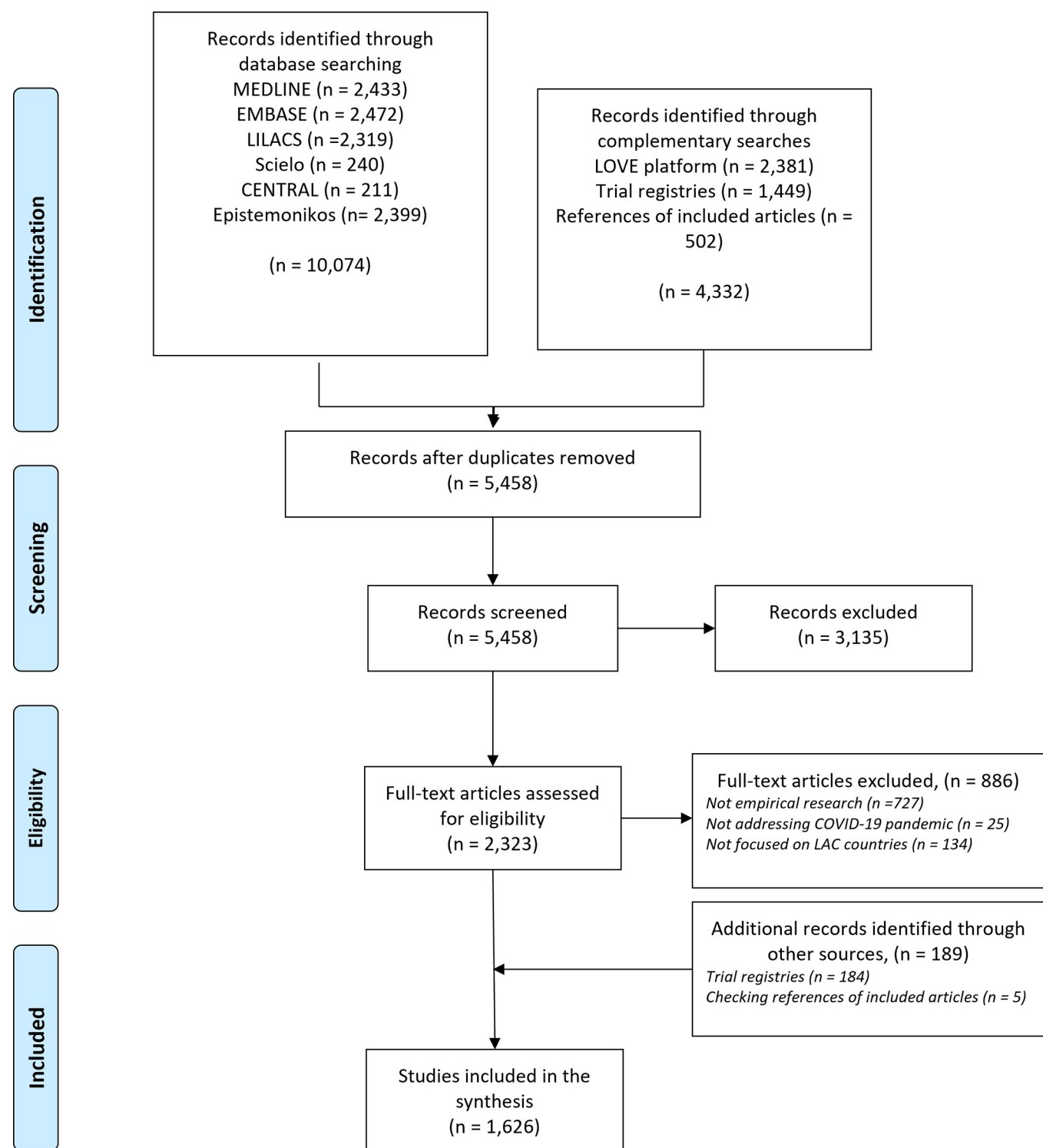

**Fig 1. Flow diagram for the scoping review of empirical research related to COVID-19 in LAC countries.**

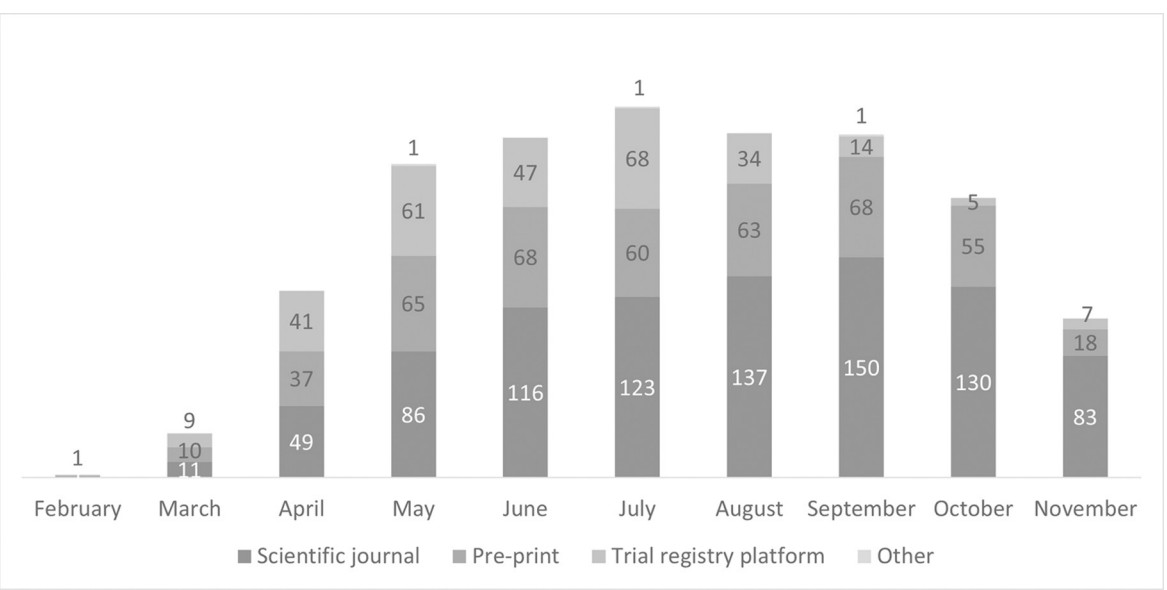

**Fig 2. Number of empirical studies published per month from February to 11 November 2020.** Note: The bar for November does not represent the total number of empirical studies published that month since the searches were conducted until November 11, 2020.

were 5 articles for which the country was not mentioned, but the authors did mention that they include at least one country from the LAC region.

Brazil concentrated most of inclusions in publications with 54.6% of the total, followed by Mexico (19.1%), Colombia (11.2%), Argentina (10.4%), Peru (10.3%), and Chile (10%), while Caribbean countries concentrated 15.3% of studies. Uruguay, Panama, Trinidad and Tobago, Chile and Costa Rica had the highest rate of empirical studies published per population with 14.9, 9.7, 9.3, 8.5 and 8 studies per 1 million population (excluding countries with less than 1 million population). Haiti and Venezuela were the only two countries with less than 2 publications per 1 million population.

179 studies (11%) were conducted in several countries including countries outside of the Region, whereas 151 were conducted in more than one country, but only in the Region.

## Data sources, methods and research questions

Most studies obtained its information directly from humans (54.7%), followed by databases (41.9%). Only 4.2% of studies used documents as data sources and 1.2% used inert sources (e.g. basic science studies). Qualitative methods were used by 4.9% of studies and 3.1% were basic sciences studies. Almost 92% of studies applied quantitative methods, with the majority of them using cross-sectional (34.7%), modelling methods (17.5%) and RCTs (13.6%). 221 studies were identified as RCTs of which 207 were ongoing studies at the time of the search. Most research questions were related to exposure, prevalence or incidence (66.4%), followed by benefits and harms of an intervention (21.5%) and prognosis (6.7%). See Table 2 for details.

## WHO Coordinated Global Research Roadmap classification of COVID-19 research

According to the modified version of the WHO Coordinated Global Research Roadmap classification of COVID19 research, 54.2% were epidemiological studies, followed by clinical management (22.3%), candidate therapeutics (12.2%), health systems arrangements (10.1%), and

**Table 2. Data sources, methods and research questions of empirical research related to COVID-19 in LAC countries.**

| | Number | Percentage of the total number of publications* |
|---|---|---|
| **Data sources** | | |
| Directly humans | 889 | 54.7% |
| Databases | 682 | 41.9% |
| Documents | 69 | 4.2% |
| Inert sources | 19 | 1.2% |
| Animals | 3 | 0.2% |
| Not described | 8 | 0.5% |
| Other | 27 | 1.7% |
| **Methodology** | | |
| Basic sciences | 50 | 3.1% |
| Quantitative | | |
| Cross-sectional study | 564 | 34.7% |
| Modelling study | 285 | 17.5% |
| Randomized controlled trial | 221 | 13.6% |
| Cohort study | 143 | 8.8% |
| Ecological study | 103 | 6.3% |
| Other** | 184 | 11.3% |
| Qualitative | 79 | 4.9% |
| Mixed-methods | 8 | 0.5% |
| **Type of research questions** | | |
| Exposure, prevalence or incidence | 1079 | 66.4% |
| Benefits and harms of an intervention | 350 | 21.5% |
| Prognosis | 109 | 6.7% |
| Views and preferences | 63 | 3.9% |
| Diagnostic accuracy | 39 | 2.4% |
| Other | 210 | 12.9% |

*% sum more than 100% because one article could have more than one category.

** Other include several types of study design such as time series, before-and-after, case-control, and economic evaluations.

infection prevention and control (8.8%). Only eleven studies were identified regarding vaccines for COVID19 and 198 studies for candidate therapeutics (see Fig 3).

## Funding sources

Almost half of papers that were included in these analyses did not report their funding source (47.5%) or did not have a specific source of funding (21.8%). Governments and the public sector contributed funding to 19.2% of papers, followed by universities and research centres with 9%. NGOs contributed to 2.4% of studies and the private sector to 4.9% (Table 3).

## Discussion

Research production in LAC during the first part of the COVID-19 pandemic surpassed 1600 empirical studies between January and November 2020. This number is difficult to compare with other regions as there are not many studies exploring the production of empirical COVID-19 research and its characteristics. To our knowledge, only the bibliometric analysis for Africa conducted by Guleid et al. 2021 [18] included "research involving the collection and

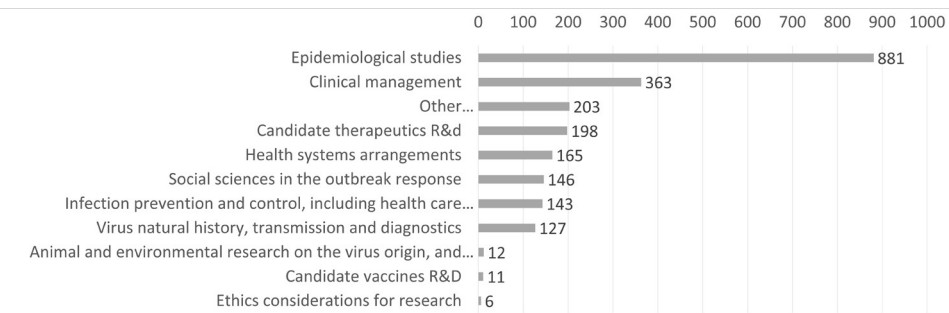

**Fig 3. Number of publications per thematic areas of empirical research related to COVID-19 in LAC countries, using the WHO COVID-19 research roadmap.**

analysis of primary data" that would be similar to our definition of "empirical research". In this case, they found 606 studies between December 2019 and December 2020, which would be close to a third of what the present review found for LAC.

Empirical COVID-19 studies were found in all LAC countries but were mainly concentrated in six countries that accounted for almost three quarters of all the included papers (Brazil, Mexico, Colombia, Peru, Chile and Argentina), while Uruguay, Panama, Trinidad and Tobago, Chile and Costa Rica had the highest rate of publications per population, with Haiti and Venezuela having the lowest. This coincides with other studies finding a similar list of countries at the top and bottom of research production among LAC countries, for instance during the COVID-19 pandemic with oncology clinical trials [19] and before the pandemic with pharmacological RCTs [20] and clinical trials [21]. The latter also found that over 80% of trials were concentrated in three countries (Brazil, Mexico and Argentina), while the present review found a larger participation of other LAC countries as well, showing that the pandemic probably has pushed for a more country widespread research generation.

Much of the research production was conducted using observational and simulation methods, with more than half of the publications classified as epidemiological studies, which correlates with the fact that two thirds of papers studied issues related to exposure, prevalence or incidence of COVID-19. Remarkably, 198 studies for candidate therapeutics and 11 studies exploring candidate vaccines were found, signaling a relatively low proportion of all empirical studies identified at the time when the present review was conducted. This is more worrying due to the fact that RCTs protocols for such studies have also been found to be of low quality

**Table 3. Funding sources of empirical research related to COVID-19 in LAC countries.**

| Funding source | Number | % of the number of publications |
|---|---|---|
| Government and public funds | 312 | 19.2% |
| Universities or research centres | 32 | 9.0% |
| Private companies | 147 | 4.9% |
| NGOs | 39 | 2.4% |
| International organizations | 79 | 2.0% |
| Public private entity | 3 | 0.2% |
| Other | 10 | 0.6% |
| Not reported | 772 | 47.5% |
| None | 355 | 21.8% |

*% sum more than 100% because one article could have more than one category.

and potential waste of research resources [15]. This finding might have implications for understanding the actual effects of different health technologies on the region's population, as most of the COVID-19 scientific knowledge would need to be imported from elsewhere. A similar situation was described in Africa, where a bibliometric analysis found that only 13 studies (1% of the total studies found) were about therapeutics and vaccines for COVID-19 [18].

Health systems research and social sciences in response to the COVID-19 pandemic had a relatively low presence, which limits the understanding of national and local realities in LAC health systems and societies, hindering decision making. On a related note, the WHO Coordinated Global Research Roadmap classification of COVID-19 research did not include an explicit category for health systems research and did not explicitly include other important issues such as mental health, which was included separately in our classification.

Regarding funding sources, close to half of studies (47.5%) did not report their funding provenance. This is higher than what has been reported in previous research, for instance related to health policy and systems research [22] and clinical trials [23] where 31% and 11% of studies, respectively, did not include funding statements. Among the studies that reported funding source, government and public sector sources were the most prevalent, while private sector sources were the least reported. This is also different from the Africa region where most of funding for COVID-19 research comes from international and foreign entities [18]. In LAC, these findings could highlight opportunities to collaborate and create synergies between public and academic funding sources, where national health research agendas could help to align priorities and efforts [24].

During the COVID-19 pandemic, some LAC countries developed programmes aiming to adapt their research production in the short term. Some examples can be found in Chile with the National Agency of Research and Development that created a special fund of USD 300,000 where researchers applied to receive grants for relevant COVID-19 research [25]; the Brazilian Ministries of Science and Technology and Health also launched public calls for COVID-19 research in 2020, totaling USD 42 million [26]; and in Argentina the National Agency for Research, Development and Innovation financed COVID-19 research projects for a total of USD 2,5 million [27]. Despite COVID-19 being a global and regional challenge, these government initiatives could have an explicit international coordination focus, which could boost synergies and expand the impact of future research in LAC.

## Policy considerations

Several LAC actors, such as governments, international organisations, universities, research centres and civil society, can use the findings of this review to understand their research production in times of a global public health emergency, which can help to identify areas of relative high research volume (e.g. epidemiologic and simulation studies) and potentially research gaps (e.g. vaccines and therapeutics) to improve collaboration between countries in the region and externally; for instance, to expand networks, to look for and pool funds, to improve surveillance systems, and to boost the production and quality of data and studies [28].

There is also a need to build capacity to have more flexible research production in order to act fast in responding to public health emergencies such as pandemics, which can be readily translated to decision making at different levels of health and social systems [29]. This seems especially relevant to research linked to existing or developing data systems. Strengthening the data infrastructure has been identified by international organizations such as OECD and WHO/PAHO as essential for managing health systems affected by a pandemic. Having a strong infrastructure analysing the R&D environment may help to better understand the

situation in the region, where multilateral collaboration among international organisations, governments and research actors can further generate this information as a public good.

This goes in line with the need for LAC to become more self-sufficient in the production of vaccines, tests, personal protective equipment, and genome sequencing [30], which can be broadened to health system performance management and research capacities, that will be key to make this happen [31]. A good example is PAHO's initiative to work with research centers in Argentina and Brazil to develop capacities to produce mRNA COVID-19 vaccines [32], along with the initiatives in Chile [33], Colombia [34] and Cuba [35] to produce COVID-19 vaccines and other technologies. A central challenge relates to the coordination of this country-level initiatives to cover the needs in the most efficient and equitable way possible. At the global level, the Multilateral Leaders Task Force on COVID-19 [36] can help with generating the much-needed synergies to deliver COVID-19 in an effective, sustainable and equitable way, but also to establish the basis for future development of capacities in LAC and other regions in the world.

## Strengths and limitations of the review

This review has several strengths. Firstly, to our knowledge, this is the first study systematically mapping and characterising all empirical research produced in response to COVID-19 in LAC. Secondly, while most of the studies exploring COVID-19 research production have had a broad bibliometric focus (e.g. counting the total number of publications, not differentiating by their empirical nature), this review only includes empirical studies using at least some basic scientific method that have a higher likelihood of contributing with novel information to literature and decision making. Thirdly, most of the studies exploring countries' involvement in COVID-19 research have looked for author's affiliation, as opposed to the actual participation of country's people or population, which is the main focus of this review. Finally, while evidence syntheses often consider certain type of study designs in their inclusion criteria, this review gathers all relevant empirical research, including all type of study designs and methods (e.g., modelling studies, basic science, etc.).

Among the limitations of this review, we could only report the month of the studies at the time the searches were conducted, which might have missed some studies that became publicly available later but with earlier dates of publication. Another limitation relates to the possibility that some studies might have not explicitly described the scientific methods used to collect and analyze the primary data, so even if they were empirical, we might have excluded them because of the review criteria. In addition, the UNESCO researcher rate was not available for all the countries, which limited the possibility of including the whole region in the publication rate per researchers (Table 1). Also, although the review did not search directly in trials registries, the databases that were searched actively search all trial registries, such as the LOVE platform and CENTRAL. To make the search even broader, the included studies of Carracedo et al. 2021 [15], which searched thoroughly in such registries, were reviewed and included when relevant. Finally, included studies were not critically appraised in this review, and therefore the quality of research cannot be assessed.

## Conclusion

During the first part of the COVID-19 pandemic, LAC countries have contributed to the global research effort primarily with epidemiological studies, with little participation on vaccines, meaning that this type of knowledge would be imported from elsewhere. All LAC countries produced COVID-19 empirical research. Research agendas could be further coordinated between different actors within and among LAC countries aiming to enhance shared self-

sufficiency to respond to the knowledge needs in the region, especially considering that most of the declared research funding came from public and academic sources.

## Supporting information

**S1 Checklist. Preferred Reporting Items for Systematic reviews and Meta-Analysis for scoping reviews (PRISMA-ScR) checklist.**
(DOCX)

**S1 File.**
(DOCX)

**S1 Data. Data extraction form.**
(XLSX)

## Acknowledgments

We acknowledge and thank Gloria Carmona for her contributions in the abstract screening process, and Dr Niek Klazinga from the Amsterdam University Medical Centre (AMC/UvA) for providing feedback on the initial findings.

## Author Contributions

**Conceptualization:** Cristián Mansilla, Cristian A. Herrera.

**Data curation:** Cristián Mansilla, Laura Boeira, Andrea Yearwood, Analia S. Lopez, Luis E. Colunga-Lozano, Eva Brocard, Tatiana Villacres, Marcela Vélez, Gabriel Di Paolantonio, Ludovic Reveiz.

**Formal analysis:** Cristián Mansilla, Cristian A. Herrera.

**Investigation:** Cristián Mansilla, Cristian A. Herrera.

**Methodology:** Cristián Mansilla, Cristian A. Herrera.

**Project administration:** Cristián Mansilla, Cristian A. Herrera.

**Supervision:** Cristián Mansilla, Cristian A. Herrera.

**Validation:** Cristián Mansilla, Cristian A. Herrera, Laura Boeira, Andrea Yearwood, Analia S. Lopez, Luis E. Colunga-Lozano, Eva Brocard, Tatiana Villacres, Marcela Vélez, Gabriel Di Paolantonio, Ludovic Reveiz.

**Writing – original draft:** Cristian A. Herrera.

**Writing – review & editing:** Cristián Mansilla, Cristian A. Herrera, Laura Boeira, Andrea Yearwood, Analia S. Lopez, Luis E. Colunga-Lozano, Eva Brocard, Tatiana Villacres, Marcela Vélez, Gabriel Di Paolantonio, Ludovic Reveiz.

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
