## [Decision Letter · Decision Letter 0]

1 Nov 2021

PONE-D-21-25411Characterising COVID-19 empirical research production in Latin America and the Caribbean: a scoping reviewPLOS ONE

Dear Dr. Herrera,

Thank you for submitting your manuscript to PLOS ONE. After careful consideration, we feel that it has merit but does not fully meet PLOS ONE’s publication criteria as it currently stands. Therefore, we invite you to submit a revised version of the manuscript that addresses the points raised during the review process. Both reviewers request a number of changes to the paper. In most cases they request clarifications and more rigorous wording. 

We look forward to receiving your revised manuscript.

Kind regards,

Alessandro Muscio

Academic Editor

PLOS ONE

The authors have declared that no competing interests exist. Authors hold sole responsibility for the views expressed in the manuscript, which may not necessarily reflect the opinion or policy of the Pan American Health Organization nor the World Bank Group.

3. We note that Figures 3a and 3b in your submission contain [map/satellite] images which may be copyrighted. All PLOS content is published under the Creative Commons Attribution License (CC BY 4.0), which means that the manuscript, images, and Supporting Information files will be freely available online, and any third party is permitted to access, download, copy, distribute, and use these materials in any way, even commercially, with proper attribution. For these reasons, we cannot publish previously copyrighted maps or satellite images created using proprietary data, such as Google software (Google Maps, Street View, and Earth). For more information, see our copyright guidelines: http://journals.plos.org/plosone/s/licenses-and-copyright.

a. You may seek permission from the original copyright holder of Figures 3a and 3b to publish the content specifically under the CC BY 4.0 license. 

In the figure caption of the copyrighted figure, please include the following text: “Reprinted from [ref] under a CC BY license, with permission from [name of publisher], original copyright [original copyright year]

4. We note that this manuscript is a systematic review or meta-analysis; our author guidelines therefore require that you use PRISMA guidance to help improve reporting quality of this type of study. Please upload copies of the completed PRISMA checklist as Supporting Information with a file name “PRISMA checklist”.

Reviewers' comments:

Reviewer's Responses to Questions

**Comments to the Author**

1. Is the manuscript technically sound, and do the data support the conclusions?

Reviewer #1: Yes

Reviewer #2: Partly

2. Has the statistical analysis been performed appropriately and rigorously? 

Reviewer #1: Yes

Reviewer #2: Yes

3. Have the authors made all data underlying the findings in their manuscript fully available?

Reviewer #1: No

Reviewer #2: Yes

4. Is the manuscript presented in an intelligible fashion and written in standard English?

Reviewer #1: Yes

Reviewer #2: Yes

5. Review Comments to the Author

Reviewer #1: This paper aims to map and analyse the existing published research related to COVID-19 by researchers with a LAC affiliation and contribute to identify opportunities for strengthening future research. This is novel and relevant research, but I have many comments relate to methods and interpretation of findings. I think the main finding that “LAC countries have contributed to the global research effort primarily with epidemiological studies, with little participation on vaccines, meaning that this type of knowledge would be imported from elsewhere.” is important. However, it needs to be contextualised with discussions about what types of infrastructures, resources and labs would be needed in LAC to produce vaccines.

Comments:

• Line 58-59: “The pandemic seems more difficult to bring under control comparing with other parts of the world, considering that the LAC region has particularly high levels of inequality and labour informality”. Compared with what? Africa? High income countries? Please elaborate as it is crucial to understand in your introduction why it is relevant to study covid-19 research in LAC.

• 108: “Articles in any language were eligible”. How did you compare topics?

• 117: “the research must be directly or explicitly connected to the current COVID-19 pandemic.” You need to explain to the reader that you used a query-based approach to find articles and also how you found the terms that you used in the query.

• 126: Brilliant that you were able to combine several bibliographic databases beyond MEDLINE. However, we need to know what is the proportion of articles from each database, and even better what is the percentage of overlap between them.

• Figure 2: Why is there a decline after July? It would make more sense to have a constant rise until November. Is there a reason related to your process of data collection?

• Figure 3: Please improve image quality. Also, “Brazil concentrated most of inclusions in publications with 54.6% of the 199 total, followed by Mexico (19.1%), Colombia (11.2%), Argentina (10.4%), Peru (10.3%), and Chile (10%), while Caribbean countries concentrated 15.3% of studies.” Would be relevant to compare these shares with shares of total medical research in LAC or total research in general. Is it possible to add all LAC research in the platforms you used and make the comparison?

• Please see also: https://unesdoc.unesco.org/in/documentViewer.xhtml?v=2.1.196&id=p%3A%3Ausmarcdef_0000377433&file=/in/rest/annotationSVC/DownloadWatermarkedAttachment/attach_import_a8477af4-1d6a-442f-af2f-7e77b02e5c31%3F_%3D377433eng.pdf&updateUrl=updateUrl7576&ark=/ark%3A/48223/pf0000377433/PDF/377433eng.pdf.multi&fullScreen=true&locale=en#page=222

• Figure 4: This figure appears in the document without any reference in the text. Although the quality of the image is very low, this is one of the most relevant analytical pieces in this article in my opinion. You need to explain how you applied “modified version of the WHO Coordinated Global Research Roadmap classification of COVID19 research” and discuss better your results.

• Table 2. It is relevant for readers need to know which of these funding sources are national, LAC or foreign. You might consider using a different kind of classification to do so: 1) LAC government/public funding; 2) Non-LAC public Funding (e.g. NIH); 3) Multilateral funding (e.g. WHO); 4) Philanthropic funding (e.g. Gates Foundation) and 5) Corporation funding. It would be interesting to know also what the biggest individual funders are and in what thematic areas they specialise.

• 247: “This number is difficult to compare with other regions as there are not many studies exploring this issue.” I am sure you will find several articles doing bibliometric analysis to COVID-19 research across the globe. Maybe not only looking at “empirical research” as you did but just using a “covid-19” query and analysing articles in PUBMED or WoS. Some examples:

• https://www.ncbi.nlm.nih.gov/pmc/articles/PMC7174863/

• https://www.scielo.br/j/spmj/a/5SyhpMcdW6RXpNnxGq88wVD/?lang=en

• 265: “Remarkably, 198 studies for candidate therapeutics and only 11 studies exploring candidate vaccines were found, signaling a relatively low participation of LAC population in this type of research.” Interesting, but compared to what? It seems a relatively high ratio when compared to Guleid et al (2021). Please find more literature that contextualizes this finding, or you need to do an international comparison to know if this is a low or high number.

• 277: “Regarding funding sources, close to half of studies (47.5%) did not report their funding provenance. This is higher than what has been reported in previous research, for instance related to health policy and systems research(21) and clinical trials(22) where 31% and 11% of studies, respectively, did not include funding statements.” I wonder if this difference is due to the type of bibliographic databases that you use to collect articles. Can you compare MEDLINE vs other databases? I might be wrong, but I think fewer studies in MEDLINE will not report funding.

• What about institutional research collaboration? What are the institutions producing more research in LAC and who are the main collaborators? Affiliation data can provide you with evidence about that.

Reviewer #2: First of all, I want to congratulate the authors of this research because COVID-19 pandemic is a very complex issue that deserves much attention and the goals of the article are both very relevant, particularly the second one ("contribute to identify opportunities for strengthening future research") and may support future decision making. I like the approach of the authors and the document is very well written and supported with references to other high quality publications.

Minor comments

1. In the section of eligibility criteria, I noticed Paraguay is missing in the list of countries. Not sure if this was a mistake or the country was excluded from the scope of the study.

2. I would suggest that tables present data sorted by some criteria. For example, table 2 could be sorted by % of the number of publications (descent)

3. line 238 says "The majority of papers ... (47.5%) ...", I would suggest to rewrite it as "Almost half of the papers ..."

4. A question about funding sources (table 2): what does "none" and "Not reported" suggest to the authors? Does it mean that people do research in their free time? Is it un-paid research? Any idea?

5. 278 "... higher than what has been reported ..." . Shouldn't it say "... higher than what has NOT been reported"?

Major comments

1. About the period of the scoping review. This study analyzes articles published between 12/2019 and 11/2020, and I understand that the cut-off date depends on the dates in which this study has been conducted. However, this pandemic has had several waves with different spreads and impacts in each country (i.e. some countries managed it well at the beginning but lost control later, others were unable to content the spread of the virus from the beginning, etc) and research communities have reacted at different speeds (some countries were faster and some ones were slower). Questions for the team: would it be possible that you extend the cut-off date of the scoping review in order to include a few more months, so you can include more research papers? and see how research evolved over time, controlling for different variables (country, research questions, methods, funding). If you can't do this, I strongly suggest that (if possible) you conduct a second study, with the same approach and data sources, for the period 12/2020 to 9/2021. This second study would help understand research focus and efforts during vaccination campaigns in LAC countries, and this would help understand other relevant questions about the pandemic.

2. 293 says "LAC countries can use the findings of this review to ...". Here, it is extremely important that the authors suggest which institutions in the countries could use these findings (i.e. Ministries of Health and Social Protection, Ministries of Science and Technology, ...who?)

3. 297-300 "There is also a need to build capacity to have more flexible research production in order to act fast in responding to public health emergencies such as pandemics, which can be readily translated to decision making at different levels of health and social systems (28)". I agree with this statement but here the key questions are: who could/should decide and change priorities about what research matters?, and where should this capacity be built? Could you elaborate on these?

4. The paper presents several very relevant findings and recommendations that may foster thinking, discussion and inform future science policies. For example, along the paper, the authors write:

in 50-51 "Research agendas could be further coordinated aiming to enhance shared self-sufficiency regarding knowledge needs in the region"

in 89-91 "research funding has also been questioned in terms of the distribution and sources for achieving a fair and sustainable research and development environment (13)"

in 91-94 "In LAC, a study searching for COVID-19 trials on treatment and prevention in the region identified “a trend towards small repetitive non-rigorous studies that duplicate efforts and drain limited resources without producing meaningful conclusions on the safety and efficacy of the interventions being tested (14)."

in 264-266 regarding a relatively very low research effort in candidate therapeutics and candidates vaccines

in 266-267 "This is more worrying due to the fact that RCTs protocols for such studies have also been found to be of low quality and potential waste of research resources (14)"

in 283-286 "In LAC, these findings could highlight opportunities to collaborate and create synergies between public and academic funding sources, where national health research agendas could help to align priorities and efforts (23)"

These six comments/findings are extremely relevant and could be the basis (or part of) to respond to the second aim of the study (contribute to identify opportunities for strengthening future research). However, I don't find a section in the paper that responds to that aim in a clear and comprehensive way, proposing recommendations for such contribution. I strongly suggest the team to elaborate on this. And it would also help an aspirational tone in such recommendations, if they want the audience to take action after reading the paper.

6. PLOS authors have the option to publish the peer review history of their article (what does this mean?). If published, this will include your full peer review and any attached files.

Reviewer #1: No

Reviewer #2: **Yes: **Rafael Anta

---

## [Author Response · Author response to Decision Letter 0]

20 Nov 2021

Dear reviewers, 

We truly thank you for your thoughtful and constructive revision of our manuscript, which we believe is now much clearer and stronger. All concerns and comments have been addressed in the revised manuscript and we have provided a response to each comment in a separate document. 

Please let us know if there is any other information missing. Many thanks.

---

## [Decision Letter · Decision Letter 1]

6 Jan 2022

PONE-D-21-25411R1Characterising COVID-19 empirical research production in Latin America and the Caribbean: a scoping reviewPLOS ONE

Dear Dr. Herrera,

Thank you for submitting your manuscript to PLOS ONE. After careful consideration, we feel that it has merit but does not fully meet PLOS ONE’s publication criteria as it currently stands. Therefore, we invite you to submit a revised version of the manuscript that addresses the points raised during the review process.

We look forward to receiving your revised manuscript.

Kind regards,

Alessandro Muscio

Academic Editor

PLOS ONE

Journal Requirements:

Additional Editor Comments:

Both reviewers appreciate your work and how you addressed their comments. One of them requires some additional minor revisions. Please follow her/his suggestions and resubmit the paper explaining how you addressed these changes. At this stage, the paper will not have to be resubmitted to the external reviewers again as I will review the changes made.

Reviewers' comments:

Reviewer's Responses to Questions

**Comments to the Author**

1. If the authors have adequately addressed your comments raised in a previous round of review and you feel that this manuscript is now acceptable for publication, you may indicate that here to bypass the “Comments to the Author” section, enter your conflict of interest statement in the “Confidential to Editor” section, and submit your "Accept" recommendation.

Reviewer #1: (No Response)

Reviewer #2: All comments have been addressed

2. Is the manuscript technically sound, and do the data support the conclusions?

Reviewer #1: Yes

Reviewer #2: Yes

3. Has the statistical analysis been performed appropriately and rigorously? 

Reviewer #1: Yes

Reviewer #2: Yes

4. Have the authors made all data underlying the findings in their manuscript fully available?

Reviewer #1: Yes

Reviewer #2: Yes

5. Is the manuscript presented in an intelligible fashion and written in standard English?

Reviewer #1: Yes

Reviewer #2: Yes

6. Review Comments to the Author

Reviewer #1: I am ok with the authors answers and changes.

However, I am unconvinced that decline of covid-19 publications from July onwards is a natural phenomenon. Please compare it with other studies in other regions to understand if this decline also appears in different regions. And please make a footnote in figure 2 saying that the month of November is incomplete.

Also, please reorder the categories of figure 3 for the categories to appear as a ranking (top category with more pubs 1st, etc.)

Reviewer #2: I read the new version of the paper, and also the responses of the authors to my comments from the previous review and I found all were addressed. Again, congratulations to the authors for such a good paper that will help promote thinking a discussion about the need for empirical research on COVID-19, and most important: how to do more and better with little resources. Coordination, collaboration and synergies will be key for research in the next future.

7. PLOS authors have the option to publish the peer review history of their article (what does this mean?). If published, this will include your full peer review and any attached files.

Reviewer #1: No

Reviewer #2: **Yes: **Rafael Anta

---

## [Author Response · Author response to Decision Letter 1]

17 Jan 2022

We thank the editor and reviewers for their new thoughtful revision of our manuscript. We respond to each of the editor and reviewer comments below, indicating where changes to the text and tables/figures can be found. 

Journal Requirements:

Author's response: We have checked our reference list and there is no retracted article.

Additional Editor Comments:

Both reviewers appreciate your work and how you addressed their comments. One of them requires some additional minor revisions. Please follow her/his suggestions and resubmit the paper explaining how you addressed these changes. At this stage, the paper will not have to be resubmitted to the external reviewers again as I will review the changes made.

Author's response: Many thanks. We provide responses to each comment below.

Reviewers' comments:

Reviewer's Responses to Questions

Comments to the Author

1. If the authors have adequately addressed your comments raised in a previous round of review and you feel that this manuscript is now acceptable for publication, you may indicate that here to bypass the “Comments to the Author” section, enter your conflict of interest statement in the “Confidential to Editor” section, and submit your "Accept" recommendation.

Reviewer #1: (No Response)

Reviewer #2: All comments have been addressed

Author's response: Many thanks.

2. Is the manuscript technically sound, and do the data support the conclusions?

Reviewer #1: Yes

Reviewer #2: Yes

Author's response: Many thanks.

3. Has the statistical analysis been performed appropriately and rigorously?

Reviewer #1: Yes

Reviewer #2: Yes

Author's response: Many thanks.

4. Have the authors made all data underlying the findings in their manuscript fully available?

Reviewer #1: Yes

Reviewer #2: Yes

Author's response: Many thanks.

5. Is the manuscript presented in an intelligible fashion and written in standard English?

Reviewer #1: Yes

Reviewer #2: Yes

Author's response: Many thanks.

6. Review Comments to the Author

Reviewer #1: I am ok with the authors answers and changes.

However, I am unconvinced that decline of covid-19 publications from July onwards is a natural phenomenon. Please compare it with other studies in other regions to understand if this decline also appears in different regions. And please make a footnote in figure 2 saying that the month of November is incomplete.

- Author's response: We have double checked the study we cite in the paper on research production in Africa (Guleid et al. 2021 (18)), along with other bibliometric analysis we found now on COVID19 publications in Asia (Tantengco OAG 2021) and in specific fields such as vaccines research (Ahmad T et al. 2021) and older adults research (Soytas RB 2021). Unfortunately, none of these studies makes a timeline analysis of the number of studies published during the pandemic. However, we would like to precise that we report that the total number of empirical studies published in LAC per month reached a peak of 252 in July, then remained relatively stable in 234 in August and 233 in September, and only in October it went down to 190. Since we don’t have the data for November onwards, we cannot really ascertain much about a significant decline in the number of publications. Therefore, we believe it would be better not to mention this issue in the paper as we don’t have enough information to make a well-grounded statement regarding a significant decline in the number of empirical studies being published per month from July onwards. 

We have added the suggested footnote in Figure 2 to make it more clear for the reader. 

Also, please reorder the categories of figure 3 for the categories to appear as a ranking (top category with more pubs 1st, etc.)

- Author's response: We have modified Figure 3 as suggested by Reviewer #1. Many thanks for this comment.

Reviewer #2: I read the new version of the paper, and also the responses of the authors to my comments from the previous review and I found all were addressed. Again, congratulations to the authors for such a good paper that will help promote thinking a discussion about the need for empirical research on COVID-19, and most important: how to do more and better with little resources. Coordination, collaboration and synergies will be key for research in the next future.

- Author's response: Many thanks. We fully agree with the comments of Reviewer #2 and hope our paper can contribute along the lines mentioned.

7. PLOS authors have the option to publish the peer review history of their article (what does this mean?). If published, this will include your full peer review and any attached files.

Do you want your identity to be public for this peer review? For information about this choice, including consent withdrawal, please see our Privacy Policy.

Reviewer #1: No

Reviewer #2: Yes: Rafael Anta

---

## [Editor Report · Decision Letter 2]

2 Feb 2022

Characterising COVID-19 empirical research production in Latin America and the Caribbean: a scoping review

PONE-D-21-25411R2

Dear Dr. Herrera,

We’re pleased to inform you that your manuscript has been judged scientifically suitable for publication and will be formally accepted for publication once it meets all outstanding technical requirements.

Kind regards,

Alessandro Muscio

Academic Editor

PLOS ONE
---

## [Editor Report · Acceptance letter]

8 Feb 2022

PONE-D-21-25411R2 

Characterising COVID-19 empirical research production in Latin America and the Caribbean: a scoping review 

Dear Dr. Herrera:

I'm pleased to inform you that your manuscript has been deemed suitable for publication in PLOS ONE. Congratulations! Your manuscript is now with our production department. 

Kind regards, 

on behalf of

Dr. Alessandro Muscio 

Academic Editor

PLOS ONE